# Low-Molecular-Weight Collagen Peptide Improves Skin Dehydration and Barrier Dysfunction in Human Dermal Fibrosis Cells and UVB-Exposed SKH-1 Hairless Mice

**DOI:** 10.3390/ijms26136427

**Published:** 2025-07-03

**Authors:** Eunjung Choi, Heeyeon Joo, Myunghee Kim, Do-Un Kim, Hee-Chul Chung, Jae Gon Kim

**Affiliations:** 1Health Food Research and Development, NEWTREE Co., Ltd., Seoul 05604, Republic of Korea; ejchoi1@inewtree.com (E.C.); hyjoo@inewtree.com (H.J.); mhkim@inewtree.com (M.K.); dkim@inewtree.com (D.-U.K.); 2INVIVO Co., Ltd., 121, Deahak-ro, Nonsan 32992, Republic of Korea

**Keywords:** UVB, collagen, hydration, TEWL, wrinkle, erythema, inflammation

## Abstract

Ultraviolet B (UVB), a component of solar ultraviolet light, is a major contributor to skin photodamage. UVB exposure primarily affects the epidermis, which leads to wrinkle formation, loss of skin elasticity, oxidative stress, and inflammation. Prolonged or intense UVB exposure can increase the risk of skin cancer. Collagen peptides are known as functional foods that improve skin dryness and wound healing. In this study, we aimed to investigate the protective and ameliorative effects of a low-molecular-weight collagen peptide (LMWCP) with a high absorption rate and photodamage. In vitro analysis using human dermal fibroblasts (HDFs) demonstrated that LMWCP promoted skin protection by increasing procollagen type I production, enhancing cell proliferation and migration, and inhibiting MMP-1 activity. Furthermore, LMWCP intake was indicated by improved skin hydration, reduced trans-epidermal water loss (TEWL), and changes in the clinical parameters, including skin elasticity, erythema, and scaling scores in UVB-exposed hairless mice. In the UVB-damaged tissues, an increase in skin elasticity-related enzymes was observed along with a decrease in aging-related and pro-inflammatory gene expression. Histological analysis revealed an increase in collagen content and restoration of dermal thickness. These findings suggested that LMWCP has significant benefits in preventing and improving UVB-induced skin damage.

## 1. Introduction

The skin is a large organ that regulates temperature, detects moisture, senses pain, fever, and other stimuli, and protects against external pathogens [1]. It is divided into epidermal, dermal, and subcutaneous fat layers [2]. The epidermis is the outermost layer and serves as the primary defense against infection, whereas the dermis, located beneath the epidermis, contains blood vessels and nerves that support and nourish [3,4,5]. The subcutaneous fat layer is responsible for maintaining the skin thickness. Fibroblasts are present in the dermis, where they produce proteins such as collagen, elastin, and mucopolysaccharides to form the extracellular matrix (ECM) [6]. Deterioration of skin function can be caused by various factors, including genetics, aging, and environmental influences [7,8]. Skin aging is divided into endogenous aging, in which the functions of all of the organs in the body decline with age, leading to changes in the structure and physiological functions of the skin, and photodamage, which is caused by oxidative stress from factors such as ultraviolet rays and environmental conditions [9]. Both aging and photodamage degrade the extracellular matrix in the dermis, thus resulting in a loss of elasticity and firmness, leading to wrinkles and dry skin owing to a decrease in mucopolysaccharides [10]. Ultraviolet (UV) B-induced skin damage causes oxidative stress, inflammation, collagen degradation, and loss of skin elasticity, leading to wrinkles and dryness [7]. Recent studies have shown that skin is not only a physical barrier but also interacts with the neuro-immuno-endocrine system to maintain homeostasis [8,9]. UVB exposure can activate inflammatory responses and disrupt collagen synthesis via neuro-immuno-endocrine pathways, contributing to further skin damage.

Collagen comprises approximately 30% of all proteins in mammals and contains 90% or more of the weight of dermal tissue [10]. Therefore, it is important to maintain skin structure and elasticity. Recent studies have confirmed that collagen intake can mitigate the increase in UV-induced wrinkles, the reduction in elasticity, and the abnormal skin thickening caused by keratinocyte proliferation [11,12,13]. Collagen peptides (CPs) have been shown to protect against UV-induced skin damage and help maintain normal skin function [14,15]. UVB exposure can activate inflammatory responses and disrupt collagen synthesis via neuro-immuno-endocrine pathways, contributing to further skin damage [9]. Therefore, CPs have emerged as a promising solution to mitigate UVB-induced skin damage by enhancing collagen production, improving hydration, and increasing skin elasticity. It can be categorized into high-molecular-weight (HMWCP; MW > 10 kDa) and low-molecular-weight (LMWCP; MW 1–5 kDa) types based on their molecular weight [16]. HMWCPs, due to their larger molecular size, exhibit a lower absorption rate in the body. These are typically used in joint supplements or muscle recovery products. On the other hand, LMWCPs, which have a smaller molecular size, are more readily absorbed by the body. It is primarily utilized for improving skin elasticity, moisture retention, and wrinkle reduction [17,18,19]. LMWCPs are typically composed of specific sequences such as Glycine-Proline-Hydroxyproline (Gly-Pro-Hyp; GPH) and Glycine-Proline-Alanine (Gly-Pro-Ala; GPA) [17]. These two major types of collagen-derived peptides have been identified for functional roles in skin health. GPH has been reported to stimulate collagen synthesis in the body, thereby enhancing skin hydration and elasticity [20]. In contrast, GPA is particularly associated with the maintenance of skin firmness and elasticity [21]. These peptides are considered key contributors to skin health and are widely studied in the context of nutraceuticals and functional foods. Recently, the intake of LMWCP has been reported to be effective in repairing or protecting dermal skin damage caused by photodamage and in improving skin hydration in human dermal fibroblasts (HDFs) [22].

In this study, we aim to evaluate the efficacy of LMWCP containing GPH and GPA in HDFs and ultraviolet B (UVB)-exposed hairless mice. By assessing effects on collagen synthesis, skin hydration, and elasticity, this study provides scientific evidence for the protective and restorative potential of LMWCP against photodamage-related skin.

## 2. Results

### 2.1. Cell Viability, Procollagen Type I Peptide Level and MMP-1 Activity in HDFs

In HDFs, the LMWCPs showed no cytotoxicity at concentrations up to 1 mg/mL, as determined by the MTT assay (Figure 1A). It enhanced collagen synthesis by increasing procollagen type I peptide (PIP) levels (Figure 1B), induced a concentration-dependent increase in cell proliferation, as determined by the BrdU assay (Figure 1C), and reduced Ultraviolet B (UVB; 10 mJ/cm^2^)-induced MMP-1 activity (Figure 1D). These results indicate that the skin structure can be improved by promoting collagen production and protecting against UVB-induced damage (Figure 1).

### 2.2. Cell Migration in HDFs

Collagen is naturally synthesized by fibroblasts in the dermis and it is known to have an affect on cell adhesion, differentiation, and migration in the skin [23,24]. Therefore, the proliferation and migration of LMWCP-treated HDFs were analyzed. As shown in Figure 1E, cell migration was assessed after treatment with 0.5 and 1 mg/mL LMWCP for 24 h. The images captured before and after treatment demonstrated a significant enhancement in cell migration following treatment with 1 mg/mL LMWCP (Figure 1E, left). It was assessed by the 0 h point as the baseline, and cell migration (%) was analyzed after 24 h (Figure 1E, right). LMWCP 0.5 did not result in a significant change in cell migration compared to the CLT (CTL: 20.58 ± 2.59% vs. LMWCP 0.5: 24.20 ± 3.14%). However, LMWCP 1 (38.49 ± 2.89%) significantly increased compared to the CTL. These findings indicated that LMWCP (1 mg/mL) promoted cell migration in HDFs.

### 2.3. LMWCP Improved the Condition of Skin on UVB-Exposed SKH-1 Hairless Mice

To investigate the effect of LMWCP on UVB-induced skin photodamage, SKH-1 hairless mice were exposed to UVB three times a week, and 200 and 400 mg/kg LMWCPs were orally administered daily during UVB exposure. For 12 weeks, skin hydration, trans epidermal water loss (TEWL), erythema, elasticity, erythema score and scaling score were measured (Table 1 and Table 2). After 8 weeks, the intake of LMWCP significantly increased skin hydration and reduced TEWL in the skin damaged by UVB exposure (Table 1). UVB-induced photodamage led to a reduction in elasticity and increased both erythema and scaling scores (Table 2). The LMWCP resulted in significant improvements in regard to the elasticity, erythema, and scaling scores. As shown in Figure 2, surface evaluation of the skin parameters Ra, Rmax, and R3z was performed in hairless mice. UVB exposure significantly increased the wrinkle parameters Ra, Rmax, and R3z, indicating that UVB irradiation induces skin surface roughening. The intake of LMWCP resulted in an improvement in skin wrinkling after 4 weeks (Figure 2B), 8 weeks (Figure 2C) and 12 weeks (Figure 2D). After 8 weeks, all parameters returned to normal levels, suggesting that LMWCP effectively ameliorated UVB-induced skin damage. Exposure to UVB significantly induced the clinical deterioration of the dorsal skin of mice, with dryness, wrinkles, erosion, and excoriation, whereas LMWCP treatment effectively reduced these parameters (Figure 2). These results suggested that LMWCP ameliorated UVB-induced skin dehydration, TEWL, elasticity, erythema score, scaling score, and wrinkles.

### 2.4. LMWCP Recovered Skin Barrier Function and Structure in SKH-1 Hairless Mice

Previous studies have demonstrated that UVB exposure induces significant skin damage (Table 1 and Table 2, Figure 2). Therefore, the levels of procollagen type I, elastin, and hyaluronic acid, which are indicators of skin structure, elasticity, and hydration, were assessed in the skin tissues of all of the experimental groups (Figure 3A–C). UVB exposure significantly decreased the levels of procollagen type I, elastin, and hyaluronic acid, indicating substantial structural and functional damage to the UVB-exposed skin. LMWCP significantly increased procollagen type I and elastin levels. These findings suggest that LMWCP positively influence skin elasticity and structure during UVB-induced damage. The skin barrier function was determined by mRNA expression levels of *Mmps* (including *Mmp-1α*, *-2*, *-9*, and *-13*), and hyaluronan synthase-1 (*Has-1*) in the skin. LMWCP significantly decreased the mRNA levels of *Mmps*, which were increased by UVB exposure (Figure 3D–G). Conversely, it significantly increased the mRNA levels of *Has-1*, which decreased upon UVB exposure (Figure 3H). These results showed that LMWCP promoted skin regeneration and protection by inhibiting skin-degrading enzymes and *Mmp* gene expression/increasing hyaluronan synthase and *Has-1* gene expression.

### 2.5. LMWCP Suppresses UVB-Induced Inflammation in SKH-1 Hairless Mice

UVB exposure induces severe skin inflammation, which occurs through various mechanisms [25]. Previous studies have shown that UVB exposure leads to the generation of reactive oxygen species, activation of inflammatory pathways, and also the release of pro-inflammatory cytokines such as IL-1β, IL-6, and TNF-α [26,27]. Therefore, the expression level of mRNA cytokines (*Il-1α*, *-1β*, -*6*, -*10*, and *Tnf-α*) and immune-related proteins (phosphorylation of MAPKs, c-Fos, and c-Jun) were investigated in the skin tissues from hairless mice (Figure 4 and Figure 5). The mRNA expression levels of pro-inflammatory cytokines and the phosphorylation of MAPKs, c-Jun, and c-Fos, indicating the activation of signaling pathways involved in cellular stress responses, inflammation, and potential skin damage, increased upon exposure to UVB radiation (Figure 4 and Figure 5). LMWCP intake inhibited the expression of inflammatory cytokines and activation of the MAPK/c-Jun/c-Fos signaling pathway induced by UVB exposure. LMWCP decreased the UVB-induced upregulation of mRNA cytokines (*Il-1α*, *-1β*, -*6*, -*10*, and *Tnf-α*) and the phosphorylation of MAPK/c-Jun/c-Fos, suggesting that collagen helps modulate the inflammatory response and signaling pathways, thus promoting skin recovery and reducing UVB-induced damage.

### 2.6. LMWCPs Improve UVB-Induced Epidermal Structural Changes and Collagen Enhancement in SKH-1 Hairless Mice

UVB-induced skin damage is characterized by increased epidermal thickness and tissue atrophy due to collagen degradation [28]. Therefore, H&E staining and IHC were performed for histological analysis (Figure 6). Our results showed that collagen type I expression in the epidermis, as assessed by IHC, was significantly reduced by UVB exposure. LMWCP intake was found to have significantly increased collagen type I expression in the epidermis (Figure 6A). Additionally, H&E staining revealed a significant increase in epidermal thickness following UVB exposure, which was restored to normal levels by LMWCP treatment.

## 3. Discussion

The skin serves as a protective barrier against various harmful external agents [1]. Among these, UVB exposure is known to induce functional deterioration and structural alterations of the skin, leading to both inflammation and accelerated aging [29]. Previous studies have reported that UVB exposure causes changes in the epidermis, such as dryness and thickening of the stratum corneum, resulting in increased epidermal thickness and impaired moisture retention [30,31]. Sunscreens are commonly used in order to reduce UVB-induced skin damage [7]. In addition, the intake of collagen supplements, vitamins, polyphenols, and flavonoids has been shown to delay UVB photodamage owing to their antioxidant and anti-inflammatory properties [32,33,34]. In this study, LMWCPs were used to investigate skin improvement following UVB-induced photodamage. The results showed that LMWCPs increased the level of PIP, cell proliferation, and migration; decreased the level of MMP-1 activity in HFDs; and ameliorated barrier dysfunction and inflammation in UVB-exposed hairless mice.

UVB can cause skin inflammation and contribute to collagen disruption, leading to photodamage [29]. Exposure to UVB not only accelerates collagen degradation but it may also impair collagen synthesis by reducing the production of procollagen, a collagen precursor [35]. The skin barrier plays an important role in retaining moisture and maintaining the structure and function of the skin [1]. However, aging and other environmental factors can weaken this barrier [36,37]. UVB exposure damages the epidermis, reduces skin hydration, and increases TEWL, ultimately impairing the function of skin barrier [38,39]. Additionally, it induces an inflammatory response in the skin, promotes oxidative stress, and accelerates collagen degradation, leading to skin damage [26,39]. These interactions within the neuro-immune-endocrine system have a significant impact on skin health. It activates various signaling pathways, including MAPKs and NF-κB, leading to the expression of inflammatory molecules and the production of *Mmps* [40,41]. These *Mmps* play major roles in collagen degradation during inflammation [40,41,42]. Furthermore, skin plays a crucial role in hormone synthesis and immune regulation through its endocrine function [43]. The TGF-β signaling pathway is particularly important in regulating the synthesis of structural proteins, such as collagen; UVB exposure can activate inflammatory responses and disrupt collagen synthesis via neuro-immuno-endocrine pathways, contributing to further skin damage [44]. CPs have emerged as a promising solution to mitigate UVB-induced skin damage by enhancing collagen production, improving hydration, and increasing skin elasticity. These mechanisms help restore the skin’s endocrine function and strengthen the skin barrier, thereby improving damage caused by UVB exposure [45,46]. Understanding these mechanisms is essential for developing strategies to protect the skin from UVB-induced damage and improve skin health. In this study, we have demonstrated that LMWCPs significantly promoted procollagen production, cell proliferation, and migration in a dose-dependent manner at non-toxic concentrations. It also inhibits MMP-1 activity in HDFs. In UVB-exposed hairless mice, LMWCP intake enhanced skin hydration and reduced TEWL. In addition, it alleviated UVB-induced erythema, elasticity, and wrinkles in the dorsal skin. LMWCP intake increased the levels of key enzymes, including procollagen type I and elastin, and enhanced mRNA expression of *Has-1*, both of which are involved in maintaining skin hydration and tissue structure [47]. Additionally, it significantly inhibited the gene expression of *Mmps* (*Mmp-1*, *Mmp-2*, *Mmp-9*, and *Mmp-13*) in skin tissue. These findings suggest that LMWCPs exert protective and therapeutic effects against UVB-induced skin damage by reducing factors associated with skin elasticity loss and wrinkle formation while enhancing the expression of genes involved in skin hydration maintenance. According to previous studies, our results demonstrate that LMWCPs inhibit UVB-induced inflammation by reducing the expression of inflammation-related cytokines (including *Il-1α*, *Il-6*, *Il-10*, and *Tnf-α*) and modulating neuro-immune-endocrine pathways through the suppression of MAPKs/c-Jun/c-Fos phosphorylation [9,29]. Furthermore, LMWCPs promote cellular regeneration and collagen synthesis in the skin, aiding in the recovery of skin damage caused by UVB exposure. The skin is a critical regulatory hub within the neuro-immune-endocrine system, playing a significant role in overall systemic health [48]. Environmental stimuli such as UVB can influence systemic health through the skin’s endocrine and immune responses [8,9,49]. LMWCPs improve skin inflammation and collagen degradation, strengthen the skin barrier function, and contribute to the health of the skin. These findings highlight the skin’s pivotal role in maintaining homeostasis both locally and systemically. These results suggest that LMWCP supplementation promotes skin regeneration at the cellular level, alleviates UVB-induced skin damage in animal models, and modulates inflammatory signaling pathways. This demonstrates its potential as a functional ingredient for improving skin health.

Collagen is the primary protein in skin that provides structure, elasticity, and firmness [50]. It is a key component of the dermis, the middle layer of the skin, where it forms fibrous networks that provide skin strength and flexibility [51]. With age, collagen production naturally declines, leading to signs of aging, such as wrinkles and sagging skin [36,46,52]. Maintaining collagen levels is important for maintaining healthy, youthful-looking skin [53]. Our results showed that LMWCP intake increased collagen type I protein levels in the epidermal tissue. These results therefore suggest that LMWCP intake alleviates UVB-induced skin damage by enhancing skin elasticity and reducing wrinkle formation. These findings are similar to those of previous studies, suggesting that LMWCPs promote collagen synthesis, resulting in improvements in skin texture, elasticity, wrinkles, and moisture retention [54,55,56]. Moreover, the distribution of the collagen peptides to the skin likely enhances their effectiveness in regard to counteracting environmental stressors like UVB exposure [14,15]. Overall, LMWCPs offer promising benefits for skin health by increasing collagen content, which contributes to both epidermal layers of the skin, improving its texture, elasticity, and skin hydration.

In conclusion, LMWCPs significantly increased PIP levels and inhibited MMP-1 activity, thereby enhancing the proliferation and migration of HDFs. In UVB-exposed hairless mice, its improved skin hydration and elasticity and reduced TEWL. Notably, significant improvements in elasticity, erythema, and scaling were observed after four weeks of oral administration. It also alleviates skin inflammation by reducing UVB-induced pro-inflammatory cytokines (*Il-1α*, *-6*, *-10*, and *Tnf-α*) and inhibiting the phosphorylation of MAPKs, c-Jun, and c-Fos. Additionally, it promotes recovery from UVB-induced skin damage by downregulating the expression of *Mmps* (*Mmp-1*, *-2*, *-9*, and *-13*) and upregulating *Has-1*, thereby enhancing skin barrier function. Lastly, despite the promising effects of LMWCPs containing GPH and GPA on skin barrier function, hydration, and inflammation, there are some limitations to this study. While the involvement of neuro-immune-endocrine interactions such as the regulation of inflammatory cytokines and MAPK/c-Jun/c-Fos suggests a role in skin recovery, the precise mechanisms by which LMWCPs influence neuroendocrine signaling have not been directly investigated. Further research is needed to elucidate these mechanisms.

## 4. Materials and Methods

### 4.1. Preparation of Low-Molecular-Weight Collagen Peptides (LMWCPs)

We used LMWCPs supplied by NEWTREE Co., Ltd. (Seoul, South Korea), which is a collagen hydrolysate derived from the skin of *Pangasius hypothalamus*. Compositional analysis by high-performance liquid chromatography confirmed the presence of GPH (>3%) and GPA (>1%), with total tripeptides accounting for more than tripeptide 15% (*w*/*w*). The LMWCPs were dissolved in distilled water.

### 4.2. Cell Culture and Cell Viability on Human Derma Fibroblasts Cells (HDFs)

Primary HDFs were purchased from American Type Culture Collection (PCS-201-012, Manassas, VA, USA) and cultured in Dulbecco’s modified Eagle medium/nutrient mixture F-12 (DMEM/F12) supplemented with 10% fetal bovine serum (FBS; 16000-044, Gibco, New York, NY, USA) and 1% antibiotic-antimycotic (15240062, ThermoFisher, Waltham, MA, USA). Culture conditions were 37 °C in a humidified atmosphere containing 5% CO_2_ (HERA cell 150i, Thermo Scientific, Waltham, MA, USA).

Cell viability was evaluated using the 3-[4,5-dimethylthiazol-2-yl]-2,5 diphenyl tetrazolium bromide (MTT) assay 24 h after LMWCP treatment in serum-free DMEM/F12. Following MTT incubation, formazan crystals were dissolved in dimethyl sulfoxide and absorbance was measured at 540 nm using an Epoch2 spectrophotometer (BioTek, Winooski, VT, USA).

### 4.3. Collagen Production and MMP-1 Activity Inhibition in HDFs

HDFs were treated with LMWCPs and TGF-β1 for 24 hrs in order to assess collagen production, and with LMWCPs and retinoic acid after UVB exposure in order to measure MMP-1 levels. The supernatants were analyzed using ELISA kits for MMP-1 (EHMMP1; Invitrogen, Miami, OK, USA) and Procollagen Type I C-Peptide (PIP; (MK101; Takara, Shiga, Japan)).

### 4.4. Proliferation and Migration of HDFs

Cell proliferation was assessed using the BrdU ELISA kit (Epoch, BioTek, Winooski, VT, USA). HDFs (1 × 10^4^ cells/well) were pre-treated with 5 µg/mL Mitomycin C for 1 h, then incubated with LMWCPs in serum-free DMEM/F12 for 24 h. The absorbance was measured at 450 nm.

Cell migration was evaluated using a scratch assay with Ibidi inserts (ibidi, GmbH, Martinsried/Munich, Germany). HDFs (7 × 10^3^ cells/well) were pre-treated with 5 µg/mL Mitomycin C, then treated with LMWCPs and 10 mg/mL FBS for 24 h. Cell migration was quantified by comparing images taken at 0 and 24 h using the ImageJ software, Ver.1.53k (Ver. 1.53k, NIH, Wayne Rasband, MD, USA).

### 4.5. UVB-Induced Photodamage and Skin Monitoring

Specific pathogen-free SKH-1 hairless mice (5 weeks old, female) were purchased from Oriental Bio (Seongnam, South Korea). This study was approved by the Institutional Animal Care and Use Committee of INVIVO Co., Ltd. (IV-RA-16-2411-17). All animals were raised individually and kept in a temperature-controlled room (12 h light/12 h dark cycle; temperature: 22 ± 3 °C; humidity: 45 ± 15%; and illumination: 150 lx–300 lx).

In this study, SKH-1 hairless female mice (total 40) were divided using a Z-arrangement based on BW into four different groups (control, vehicle (only UVB exposure), LMWCP 200 (UVB exposure + 200 mg/kg LMWCP), LMWCP400 (UVB exposure + 400 mg/kg LMWCP); 10 per group). The dorsal skin of the experimental animals was exposed to UVB using T-8. M UV lamp (UVB, 312 nm, 8 W). UVB was exposure 3 times per week. The exposure doses were as follows: 60 mJ/cm^2^ during the first week, 120 mJ/cm^2^ in the second week, and 180 mJ/cm^2^ in the third week. From the fourth week through the twelfth week, the dose was maintained at 240 mJ/cm^2^. The total cumulative UVB dose over the 12 weeks was 126 MED (Figure 7).

### 4.6. Skin Monitoring on SKH-1 Hairless Mice

For 12 weeks, skin erythema and keratin clinical indices were measured by referring to the Psoriasis Area and Severity Index scores (Table 3) [57,58]. Four researchers independently examined the back skin and assigned scores based on their observations. Skin hydration, TEWL, skin elasticity, and wrinkles were assessed. Skin hydration and TEWL were measured by GPSkin Barrier^®^ (GPOWER Inc., Seoul, South Korea), elasticity with a digital caliper (Mitutoyo Inc., Tokyo, Japan), and wrinkles using a silicone replica kit (Courage + Khazaka electronic GmbH, Cologne, Germany).

### 4.7. ELISA and Western Blot

The target proteins (Appendix A) were extracted from skin using T-PER™ reagent Reagent (Thermo Scientific™, Cat. no.78510, Waltham, MA, USA) and used for Western blotting and ELISA. Procollagen type I, elastin, and hyaluronic acid levels were measured in the skin of SHK-1 mice. Tissue enzyme levels were determined using an ELISA kit (MyBioSource, MBS761657, Vancouver, BC, Canada), according to the manufacturer’s instructions. Enzyme levels were measured using a Sunrise ELISA plate reader (Tecan, Mönnedorf, Switzerland).

Equal amounts of proteins were separated by sodium dodecyl sulfate-polyacrylamide gel electrophoresis, transferred to PVDF membranes, and probed with specific antibodies. Target proteins were detected by chemiluminescence (SURMODICS IVD, Inc., Eden Prairie, MN, USA) and analyzed using a c300 imaging system (Azure Biosystems, Dublin, CA, USA) equipped with an Image Lab.

### 4.8. mRNA Expression

To extract the RNA from the HDFs and skin, the AccuPrep^®^ Universal RNA Extraction Kit (K-3140, Bioneer, Daejeon, South Korea) was used, and the extracted RNA was quantified before synthesizing cDNA. mRNA expression (Appendix A) was analyzed using quantitative real-time polymerase chain reaction (qPCR).

### 4.9. H&E Stain and Immunohistochemistry

After euthanasia, the dorsal skin of hairless mice was fixed in 10% neutral-buffered formalin, embedded in paraffin, and sectioned at 3 μm thickness. The sections were stained with hematoxylin and eosin (H&E), and subjected to immunohistochemistry (IHC, Invitrogen, PA1-26204, Collagen type 1 stain). Images were scanned and analyzed using Motic EasyScan One (Kowloon Bay, Kowloon, Hong Kong) and Case Viewer (3DHISTECH Ltd., Budapest, Hungary).

### 4.10. Statistical Analysis

All data are expressed as the mean ± standard error of the mean (SEM), and the differences between groups were analyzed using one-way ANOVA (Duncan’s multiple-range test). All statistical analyses were performed using SPSS version 23.0 (SPSS Corp. Armonk, NY, USA). Each value represents the mean of at least three independent experiments for each group. Statistical significance was set at *p* < 0.05.

## Figures and Tables

**Figure 1 ijms-26-06427-f001:**
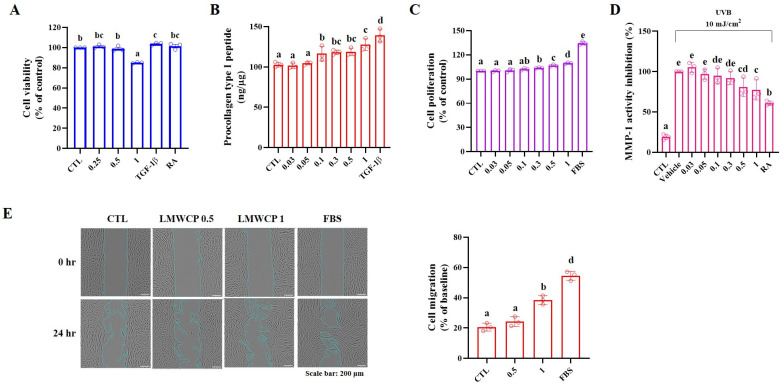
The effect of treatment of LMWCP on HDFs. (**A**), cell viability (%); (**B**), procollagen type I peptide level (ng/μg); (**C**), cell proliferation (%); (**D**), MMP-1 activity inhibition (%); (**E**), representative image (left; magnification: ×40, scale bar: 200 μm) and cell migration (right; % of baseline, 0 h). (a–e) Values in the row with different superscript letters are significantly different, *p* < 0.05; *n* = 3.

**Figure 2 ijms-26-06427-f002:**
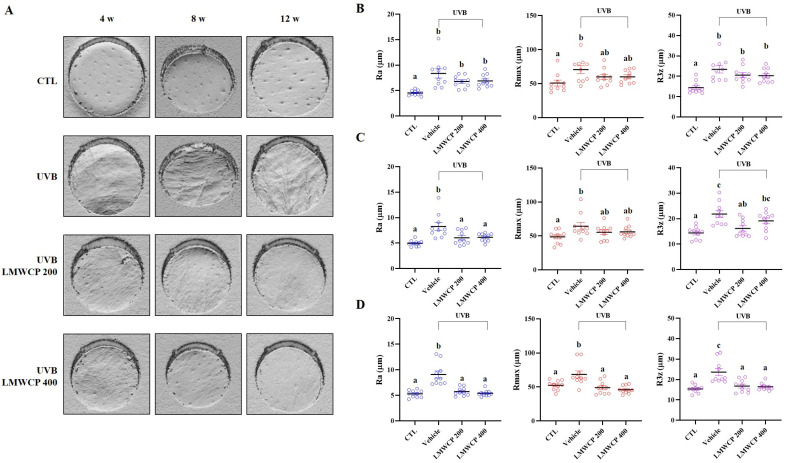
Effect of LMWCP on skin wrinkle formation in UVB-exposed SKH-1 hairless mice. The dorsal skin of the SKH-1 mice was exposed to UVB three times a week for 12 weeks. The wrinkles on replica skin were measured by Primos^CR^. The parameters for wrinkle formation, including Ra, Rmax, and R3z, were obtained from the skin replica analysis. (**A**), images of skin replicas were obtained from the dorsal skin of SKH-1 hairless mice following UVB exposure for 4 (left), 8 (middle) and 12 (right) weeks; (**B**), Ra, Rmax, and R3z of 4 weeks; (**C**), Ra, Rmax, and R3z of 8 weeks; (**D**), Ra, Rmax, and R3z of 12 weeks (*n* = 10). (a–c) Values in the row with different superscript letters are significantly different, *p* < 0.05; *n* = 10.

**Figure 3 ijms-26-06427-f003:**
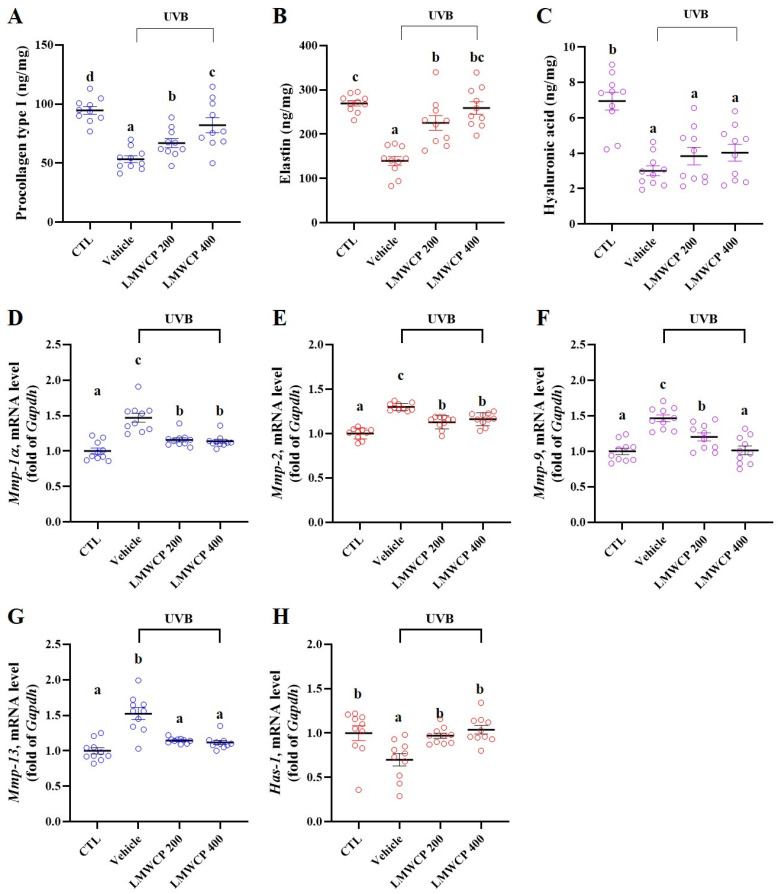
Effect of LMWCP on the levels of skin structure and hydration-related enzymes and mRNA expression in dorsal skin of UVB-exposed SKH-1 hairless mice. The dorsal skin of the SKH-1 mice was exposed to UVB three times a week for 12 weeks. (**A**), level of procollagen type I (ng/mg); (**B**), elastin (ng/mg); (**C**), hyaluronic acid (ng/mg); (**D**), mRNA expression of *Mmp-1α*; (**E**), mRNA expression of *Mmp-2*; (**F**), mRNA expression of *Mmp-9*; (**G**), mRNA expression of *Mmp-13*; and (**H**), mRNA expression of *Has-1*. (a–d) Values in the row with different superscript letters are significantly different, *p* < 0.05; *n* = 10.

**Figure 4 ijms-26-06427-f004:**
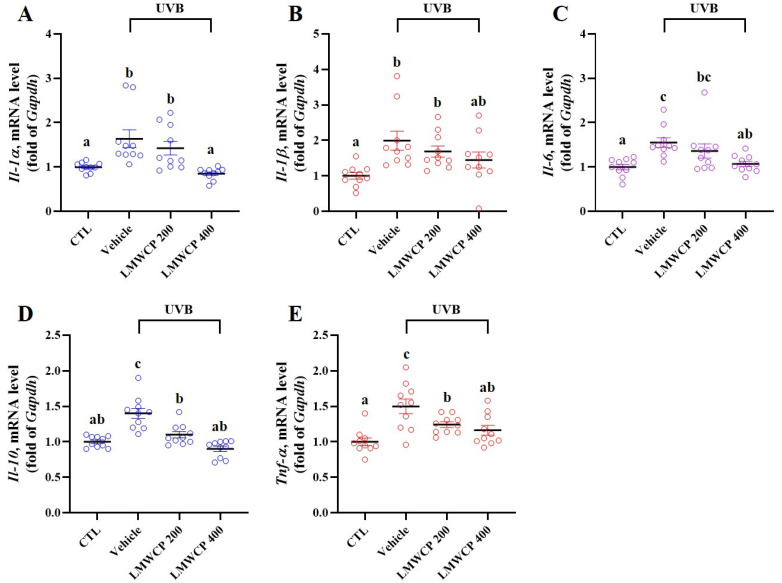
Effect of LMWCP on the expression of inflammation-related mRNA in dorsal skin of UVB-exposed SKH-1 hairless mice. The dorsal skin of the SKH-1 mice was exposed to UVB three times a week for 12 weeks. (**A**), mRNA expression of *Il-1α*; (**B**), mRNA expression of *Il-1β*; (**C**), mRNA expression of *Il-6*; (**D**), mRNA expression of *Il-10*; and (**E**), mRNA expression of *Tnf- α*. (a–c) Values in the row with different superscript letters are significantly different, *p* < 0.05; *n* = 10.

**Figure 5 ijms-26-06427-f005:**
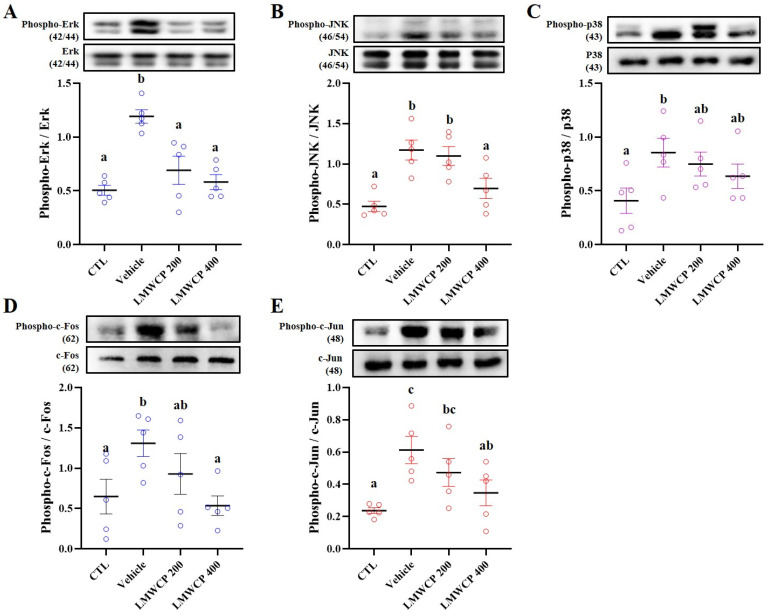
Effect of LMWCP on the MAPKs/c-Fos/c-Jun pathway in dorsal skin of UVB-exposed SKH-1 hairless mice. The dorsal skin of the SKH-1 mice was exposed to UVB three times a week for 12 weeks. Representative images of target protein (above) and target protein ratio (below). (**A**), Erk; (**B**), JNK; (**C**), p38; (**D**), c-Fos; and (**E**), c-Jun. (a–c) Values in the row with different superscript letters are significantly different, *p* < 0.05; *n* = 5.

**Figure 6 ijms-26-06427-f006:**
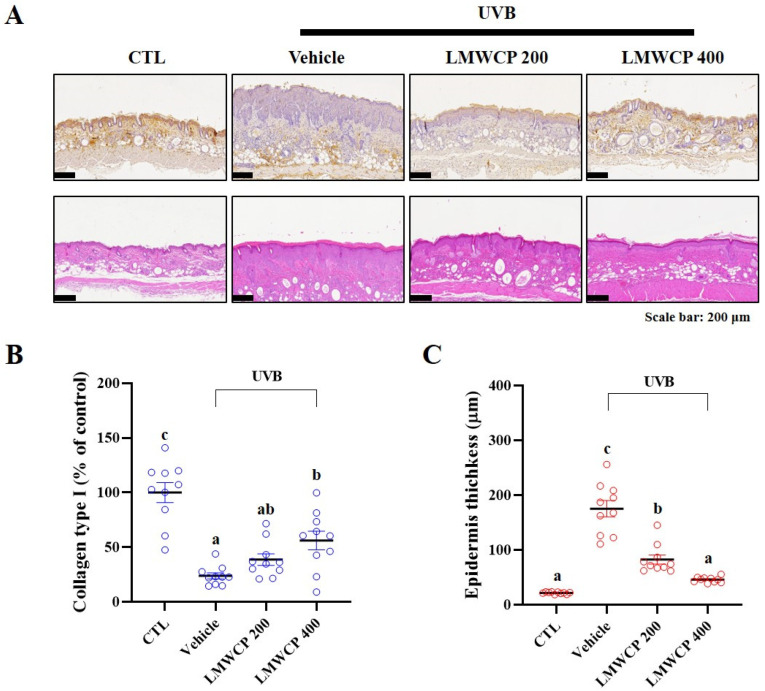
Effect of LMWCP on UVB-exposed collagen degradation and epidermis thickness in skin tissue of SKH-1 hairless mice. The dorsal skin of the SKH-1 mice was exposed to UVB three times a week for 12 weeks. (**A**), representative images, immunochemical staining of collagen I (above), H&E stain (below); (**B**), quantification of collagen I (%) on the epidermis; (**C**), epidermis thickness (μm); magnification, 10×; scale bar = 200 μm. (a–c) Values in the row with different superscript letters are significantly different, *p* < 0.05; *n* = 10.

**Figure 7 ijms-26-06427-f007:**
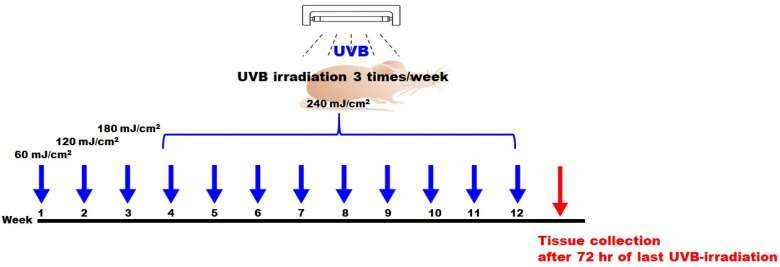
Schedule of UVB exposure on SKH-1 hairless mice.

**Table 1 ijms-26-06427-t001:** Skin hydration (A.U.) and TEWL (g/m^2^/h) of dorsal skin in UVB-exposed SKH-1 hairless mice skin on the oral administration of LMWCP for 12 weeks. (a–d) Values in the row with different superscript letters are significantly different, *p* < 0.05; *n* = 10.

	Group	0	4	8	12
Skin hydration (A.U.)	CTL	38.60 ± 0.83 ^a^	29.53 ± 1.30 ^b^	31.85 ± 1.57 ^c^	34.60 ± 1.09 ^c^
	Vehicle	37.50 ± 2.11 ^a^	16.68 ± 1.62 ^a^	10.35 ± 1.21 ^a^	10.00 ± 1.07 ^a^
	UVB LMWCP 200	39.60 ± 1.14 ^a^	16.33 ± 1.83 ^a^	14.90 ± 1.75 ^b^	15.90 ± 0.86 ^b^
	UVB LMWCP 400	41.20 ± 0.77 ^a^	19.85 ± 1.60 ^a^	16.40 ± 1.49 ^b^	15.80 ± 0.53 ^b^
TEWL (g/m^2^/h)	CTL	3.78 ± 0.29 ^a^	6.84 ± 0.35 ^a^	6.25 ± 0.41 ^a^	3.25 ± 0.33 ^a^
	Vehicle	3.80 ± 0.46 ^a^	26.03 ± 1.11 ^b^	30.60 ± 2.74 ^c^	31.40 ± 1.93 ^d^
	UVB LMWCP 200	4.05 ± 0.22 ^a^	24.07 ± 1.16 ^b^	14.50 ± 0.92 ^b^	13.55 ± 0.61 ^c^
	UVB LMWCP 400	3.30 ± 0.33 ^a^	23.79 ± 1.83 ^b^	10.60 ± 0.82 ^b^	9.80 ± 0.65 ^b^

**Table 2 ijms-26-06427-t002:** Skin elasticity, erythema sore and scaling score of dorsal skin in UVB-exposed SKH-1 hairless mice skin on the oral administration of LMWCP for 12 weeks. (a–d) Values in the row with different superscript letters are significantly different, *p* < 0.05; *n* = 10.

	Group	0	4	8	12
Elasticity (mm)	CTL	12.23 ± 0.29 ^b^	12.51 ± 0.18 ^c^	13.11 ± 0.23 ^c^	13.29 ± 0.29 ^d^
	Vehicle	11.04 ± 0.13 ^a^	9.65 ± 0.07 ^a^	8.89 ± 0.13 ^a^	7.58 ± 0.23 ^a^
	UVB LMWCP 200	10.99 ± 0.26 ^a^	10.05 ± 0.14 ^a^	9.88 ± 0.13 ^b^	9.83 ± 0.27 ^b^
	UVB LMWCP 400	11.77 ± 0.20 ^b^	10.90 ± 0.24 ^b^	10.31 ± 0.12 ^b^	10.95 ± 0.36 ^c^
Erythema Score	CTL	0.00 ± 0.00 ^a^	0.00 ± 0.00 ^a^	0.00 ± 0.00 ^a^	0.00 ± 0.00 ^a^
	Vehicle	0.00 ± 0.00 ^a^	1.70 ± 0.12 ^c^	2.38 ± 0.14 ^c^	2.40 ± 0.14 ^d^
	UVB LMWCP 200	0.00 ± 0.00 ^a^	1.73 ± 0.12 ^c^	1.18 ± 0.12 ^b^	1.65 ± 0.12 ^c^
	UVB LMWCP 400	0.00 ± 0.00 ^a^	1.37 ± 0.12 ^b^	1.08 ± 0.05 ^b^	1.18 ± 0.09 ^b^
Scaling score	CTL	0.00 ± 0.00 ^a^	0.00 ± 0.00 ^a^	0.00 ± 0.00 ^a^	0.00 ± 0.00 ^a^
	Vehicle	0.00 ± 0.00 ^a^	1.10 ± 0.07 ^b^	2.15 ± 0.20 ^c^	1.43 ± 0.12 ^c^
	UVB LMWCP 200	0.00 ± 0.00 ^a^	1.17 ± 0.07 ^b^	1.45 ± 0.12 ^b^	1.08 ± 0.04 ^b^
	UVB LMWCP 400	0.00 ± 0.00 ^a^	1.03 ± 0.03 ^b^	1.35 ± 0.15 ^b^	1.00 ± 0.00 ^b^

**Table 3 ijms-26-06427-t003:** Determination of skin erythema sore and scaling score.

Score	0	1	2	3	4
Erythema	Fleshy pink	Minor reddening Across surface	Medium red Across surface	Medium red across surface with dark red patches	Dark red
Scaling	No scaling	Slight scaling	Moderate scaling	Marked scaling	Maximal scaling

## Data Availability

Data will be made available upon request.

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
