# Peer review of "Low-Molecular-Weight Collagen Peptide Improves Skin Dehydration and Barrier Dysfunction in Human Dermal Fibrosis Cells and UVB-Exposed SKH-1 Hairless Mice"

_ijms, 2025, doi:10.3390/ijms26136427_

Round 1

Reviewer 1 Report

Comments and Suggestions for Authors

I cannot find anything novel about this paper.

Even if the collagen raw material is new, the problem is that there is no explanation for this.

The research methods also lack any novelty.

Suggestion for Improve Work: revisions or additional experiments that could make the study more compelling or
insightful.

The following points need to be clarified:
1. Detailed analysis of the research materials
2. What components of collagen peptides are effective?

Author Response

To. Reviewer 1

First of all, thank you for your valuable feedback.

The modifications have been highlighted in red for clarity.

Comments 1: Detailed analysis of the research materials

Response 1: In general, both commercial LMWCP products and materials used in previous studies contain Gly‑Pro‑Hyp (GPH) at concentrations ranging from approximately 3% to 4%. However, the specific content of Gly‑Pro‑Ala (GPA) is rarely reported. In this study, the LMWCP used was found to contain over 1% GPA, a compositional feature not commonly observed in previously reported materials. The manuscript has been revised to highlight this unique characteristic. The presence of both GPH and GPA in the material may contribute to its biological activity and provides new insights into the functional potential of collagen peptides.

Comments 2: What components of collagen peptides are effective?

Response 2: The bioactivity of collagen peptides is largely attributed to specific tripeptide sequences, such as GPH and GPA, which exhibit high bioavailability and tissue-specific effects. Compositional analysis confirmed that the LMWCP used in this study contains both GPH and GPA. These peptides are known to stimulate collagen synthesis, modulate inflammation, and enhance hydration and structural integrity in skin and connective tissues.

Reviewer 2 Report

Comments and Suggestions for Authors

The cumulative UVB dose (126 MED over 12 weeks) is inconsistently described (Section 4.5 vs. Figure 7). Stopping exposure 72 hours pre-harvest (to avoid "direct effects") may confound results. 

 Skin hydration/TEWL data are merged without clear group demarcation (e.g., "LMWCP 200" rows lack parameter labels), complicating interpretation of dose-dependent efficacy. Statistical notation (superscript letters) is misaligned with groups. 

Claims "significant enhancement" at 1 mg/mL LMWCP, yet only qualitative images (×40 magnification) are provided. No quantitative metrics (e.g., wound closure percentage) support the conclusion. 

The background on wound healing could be improved by incorporating references such as 10.1016/j.celbio.2025.100020 and 10.1002/smll.202307096. Additionally, what are the distinct advantages of the current study over similar published works in the field?

mRNA levels of MMPs/inflammatory cytokines are measured (Figures 3–5), but protein expression (e.g., MMP-1/IL-6 via Western blot/ELISA) is omitted.

In mice, LMWCP 200 mg/kg outperforms 400 mg/kg in reducing TEWL/erythema (Tables 1–2), contradicting typical dose-dependent effects. No rationale is provided for this anomaly. 

Immunohistochemistry shows collagen type I in the epidermis (Fig 6A–B), contradicting established biology where collagen I is dermal. This suggests potential antibody nonspecificity or tissue layer misidentification. 

Author Response

To. Reviewer 2

First of all, thank you for your valuable feedback.

The modifications have been highlighted in red for clarity.

Comments 1: The cumulative UVB dose (126 MED over 12 weeks) is inconsistently described (Section 4.5 vs. Figure 7). Stopping exposure 72 hours pre-harvest (to avoid "direct effects") may confound results. 

Response 1: 1. I modified it as below at “4.5 UVB-induced photodamage and skin monitoring”.

“UVB was exposure 3 times per week. The exposure doses were as follows: 60 mJ/cm² dur-ing the first week, 120 mJ/cm² in the second week, and 180 mJ/cm² in the third week. From the fourth week through the twelfth week experiment, the dose was maintained at 240 mJ/cm².”

Comments 2: Skin hydration/TEWL data are merged without clear group demarcation (e.g., "LMWCP 200" rows lack parameter labels), complicating interpretation of dose-dependent efficacy. Statistical notation (superscript letters) is misaligned with groups. 

Response 2: I modified at “Table 1”

Comments 3: Claims "significant enhancement" at 1 mg/mL LMWCP, yet only qualitative images (×40 magnification) are provided. No quantitative metrics (e.g., wound closure percentage) support the conclusion. 

Response 3: I modified at “Figure 1 legend” and “2.2. Cell migration in HDFs”.

Comments 4: The background on wound healing could be improved by incorporating references such as 10.1016/j.celbio.2025.100020 and 10.1002/smll.202307096. Additionally, what are the distinct advantages of the current study over similar published works in the field?

Response 4: The merits of this study - In general, both commercial material and previous research that LMWCP contain Gly‑Pro‑Hyp (GPH) at levels ranging from approximately 3% to 4%. However, the specific content of Gly-Pro-Ala (GPA) is not usually indicated in these sources. In our study, the LMWCP material was analyzed and confirmed to contain over 1% GPA, in addition to GPH, which is typically present at 3–4%. This rich composition allowed for a comprehensive evaluation of LMWCP's efficacy. Notably, our results showed significant improvements in skin elasticity, visible wrinkle reduction, and marked alleviation of skin inflammation and erythema—demonstrating outcomes that are not only consistent with previous research but also highlight the enhanced potential of our specific LMWCP. I provide Certificate of analysis of LMWCP and modified at “4.1, "Preparation of low-molecular collagen peptides (LMWCP)."

Comments 5: mRNA levels of MMPs/inflammatory cytokines are measured (Figures 3–5), but protein expression (e.g., MMP-1/IL-6 via Western blot/ELISA) is omitted.

Response 5: In this study, we focused on evaluating the transcriptional regulation of MMPs and inflammatory cytokines by measuring their mRNA levels using quantitative PCR, as shown in Figures 3–5. While protein-level validation through techniques such as Western blot or ELISA (e.g., for MMP-1 or IL-6) would indeed provide additional confirmation of our findings, our current approach was intended to assess gene expression changes as a first step in understanding the molecular effects of LMWCP. We agree that protein-level analysis would further strengthen our conclusions, and we plan to include such assays in future studies to provide a more comprehensive understanding of the underlying mechanisms.

Comments 6: In mice, LMWCP 200 mg/kg outperforms 400 mg/kg in reducing TEWL/erythema (Tables 1–2), contradicting typical dose-dependent effects. No rationale is provided for this anomaly. 

Response 6: The evaluation was conducted by four researchers who independently observed the samples and assigned scores individually. Since each researcher assessed the samples separately, there may have been inter-individual variability in scoring, which could have contributed to the lack of a clear dose-dependent effect in some parameters. 

Comments 7: Immunohistochemistry shows collagen type I in the epidermis (Fig 6A–B), contradicting established biology where collagen I is dermal. This suggests potential antibody non-specificity or tissue layer misidentification. 

Response 7: In our analysis, only the epidermal region was specifically selected and analyzed across all images to ensure consistency and relevance to the study.

Round 2

Reviewer 2 Report

Comments and Suggestions for Authors

accept

Author Response

I sincerely appreciate your decision to accept my manuscript.